# A multiethnic genome-wide analysis of 44,039 individuals identifies 41 new loci associated with central corneal thickness

Hélène Choquet [1✉], Ronald B. Melles[2], Jie Yin[1], Thomas J. Hoffmann [3,4], Khanh K. Thai[1], Mark N. Kvale[3], Yambazi Banda[3], Alison J. Hardcastle [5,6], Stephen J. Tuft[7], M. Maria Glymour[4], Catherine Schaefer [1], Neil Risch[1,3,4], K. Saidas Nair[8], Pirro G. Hysi [9,10,11] & Eric Jorgenson [1✉]

Central corneal thickness (CCT) is one of the most heritable human traits, with broad-sense heritability estimates ranging between 0.68 to 0.95. Despite the high heritability and numerous previous association studies, only 8.5% of CCT variance is currently explained. Here, we report the results of a multiethnic meta-analysis of available genome-wide association studies in which we find association between CCT and 98 genomic loci, of which 41 are novel. Among these loci, 20 were significantly associated with keratoconus, and one (RAPSN rs3740685) was significantly associated with glaucoma after Bonferroni correction. Two-sample Mendelian randomization analysis suggests that thinner CCT does not causally increase the risk of primary open-angle glaucoma. This large CCT study explains up to 14.2% of CCT variance and increases substantially our understanding of the etiology of CCT variation. This may open new avenues of investigation into human ocular traits and their relationship to the risk of vision disorders.

[1] Kaiser Permanente Northern California (KPNC), Division of Research, Oakland, CA 94612, USA. [2] KPNC, Department of Ophthalmology, Redwood City, CA 94063, USA. [3] Institute for Human Genetics, University of California San Francisco (UCSF), San Francisco, CA 94143, USA. [4] Department of Epidemiology and Biostatistics, UCSF, San Francisco, CA 94158, USA. [5] UCL Institute of Ophthalmology, University College London, London, UK. [6] National Institute of Health Research Biomedical Research Centre for Ophthalmology, and UCL Institute of Ophthalmology, London, UK. [7] Moorfields Eye Hospital, London, UK. [8] Departments of Ophthalmology and Anatomy, School of Medicine, UCSF, San Francisco, CA 94143, USA. [9] King's College London, Section of Ophthalmology, School of Life Course Sciences, London, UK. [10] King's College London, Department of Twin Research and Genetic Epidemiology, London, UK. [11] University College London, Great Ormond Street Hospital Institute of Child Health, London, UK. ✉email: Helene.Choquet@kp.org; Eric.Jorgenson@kp.org

Central corneal thickness (CCT) is an interesting morphological trait of cornea. Reduced CCT is a feature of keratoconus[1], and, by some accounts, is associated with an increased risk of primary open-angle glaucoma (POAG)[2–4]. Epidemiological observational studies have implicated demographic and clinical risk factors that influence CCT, including, sex, glaucoma diagnosis, and ethnicity[5–8]. Individuals of African ancestry have thinner CCT but also increased POAG risk, and have worse visual field damage and disease progression compared to other populations[5–7,9,10]. It is, however, not clear whether the observed variation in CCT confounds, or directly contributes to the greater glaucoma burden in African ancestry populations. Because of the association between CCT and the common vision disorders noted above, the identification of genes which influence CCT may open new avenues for understanding the etiology of these disorders.

CCT has a strong genetic component, with heritability estimates ranging between 0.68 and 0.95[11–13]. Recently, Iglesias et al.[14] reported 44 CCT-genomic regions in a cross-ancestry meta-analysis, including 19 novel loci awaiting independent replication. These 44 CCT-loci account for ~8.5% of the variance for this ocular trait.

Here, we present a large and ethnically diverse human genetic study of CCT, including, for the first time to our knowledge, African American and Hispanic/Latino individuals. Our study utilizes data from 44,039 individuals between the Genetic Epidemiology Research in Adult Health and Aging (GERA) cohort and the International Glaucoma Genetics Consortium (IGGC)[14]. We evaluate the effect of genetic ancestry on CCT and conduct multiethnic GWAS, identifying many novel loci. The associated loci provide candidate genes and relevant pathways and highlight differential expression of these CCT-associated genes in human ocular tissues, including cornea and lens. We also assess the effect of newly and previously identified CCT loci on the ancestry effects observed in the GERA African American ethnic group. To evaluate the clinical relevance of CCT, we examine associations of lead CCT-associated single nucleotide variations (SNVs) with keratoconus and POAG. Finally, we conduct a two-sample Mendelian Randomization analysis to clarify the nature of the relationship between CCT and POAG.

## Results

**GERA cohort and CCT.** The GERA cohort is an unselected cohort of adult members of the Kaiser Permanente Northern California integrated health care delivery system, with ongoing longitudinal records from vision examinations. For this study, our GERA sample consisted of 18,129 individuals from four ethnic groups (79.9% non-Hispanic white, 7.5% Hispanic/Latino, 8.4% East Asian, and 4.2% African American) with a measured CCT (Table 1). In our GERA sample, African Americans had thinner CCTs on average than other groups, consistent with previous reports[5–7].

**Variation in CCT by ethnicity and genetic ancestry.** To examine how CCT varied within each ethnic group, we assessed the association between the first two ancestry principal components (PCs), representing geographic origin and calculated within each group separately[15], and CCT. Within our African American sample, greater African (versus European) ancestry (PC1) was associated with thinner CCTs ($P = 3.01 \times 10^{-6}$) (Fig. 1 and Supplementary Data 1). We also observed a significant association of thicker CCT in northeastern versus northwestern European ancestry (PC2, $P = 1.20 \times 10^{-5}$) within our non-Hispanic white sample.

**GWAS of CCT in GERA.** We conducted a GWAS meta-analysis of CCT in GERA, combining results from four individual ethnic groups (Supplementary Figs. 1–5), and replicated 34 (out of the 44 loci previously reported[14]) at a Bonferroni significance level ($P \leq 1.14 \times 10^{-3}$, 0.05/44) (Supplementary Data 2). Further, five additional SNPs were nominally significant ($P < 0.05$). The effect estimates of these 39 (34 + 5) loci were in the same direction as in the IGGC cross-ancestry meta-analysis. Lead SNPs at the remaining five loci (i.e. COL8A2, ADAMTS2, SAMD9, LOXL2, and COL6A2) did not reach nominal significance in the GERA meta-analysis. The 28 loci that reached genome-wide significance in the GERA meta-analysis, 24 replicated at a Bonferroni level of significance ($P \leq 1.79 \times 10^{-3}$, 0.05/28) in either the IGGC cross-ancestry meta-analysis or the IGGC European-specific meta-analysis (Supplementary Data 3). Further three additional SNPs were nominally significant ($P < 0.05$). Only the lead SNP rs112024264 at the novel locus ZNF680 did not reach nominal significance in IGGC and so was not validated.

**Multiethnic meta-analysis of GERA and IGGC.** We then conducted a meta-analysis of CCT combining results from GERA and IGGC. This combined meta-analysis identified 74 loci associated with CCT ($P < 5 \times 10^{-8}$), of which 31 were novel (Fig. 2, Table 2, Supplementary Fig. 6, and Supplementary Data 4). The effect estimates of the 31 lead SNPs at novel loci were consistent across the 2 studies (Fig. 3), and no significant heterogeneity was observed between them (Table 2). Conducting a GWAS meta-analysis of European-specific cohorts (GERA + IGGC Europeans only) and a GWAS meta-analysis of Asian-specific cohorts

**Table 1 Characteristics of GERA subjects included in the current study by sex, and ethnic group.**

| | | CCT sample | | POAG | | Keratoconus | |
|---|---|---|---|---|---|---|---|
| | | N | Mean CCT (od,os) (μm) Mean ± SD | Cases | Controls | Cases | Controls |
| N | | 18,129 | 546.2 ± 34.9 | 4986 | 58,426 | 207 | 97,375 |
| Age at specimen (years) | Mean ± SD | 68.5 ± 10.0 | — | 59.7 ± 13.7 | 62.7 ± 13.7 | 59.7 ± 13.7 | 62.7 ± 13.7 |
| Sex | Female | 10,655 | 545.7 ± 34.2 | 2689 | 35,193 | 112 | 57,525 |
| | Male | 7474 | 547.0 ± 35.8 | 2297 | 23,233 | 95 | 39,850 |
| Ethnicity | NHW | 14,489 | 547.8 ± 34.7 | 3836 | 48,065 | 157 | 78,426 |
| | H/L | 1367 | 541.3 ± 34.8 | 411 | 4778 | 29 | 8461 |
| | EAS | 1516 | 543.4 ± 33.9 | 441 | 4034 | 8 | 7345 |
| | AA | 757 | 530.6 ± 35.5 | 298 | 1549 | 13 | 3143 |

*N* number of participants, *SD* standard deviation, *od* right eye, *os* left eye, *NHW* non-Hispanic whites, *H/L* Hispanic/Latinos, *EAS* East Asians, *AA* African-Americans, *POAG* primary open-angle glaucoma.

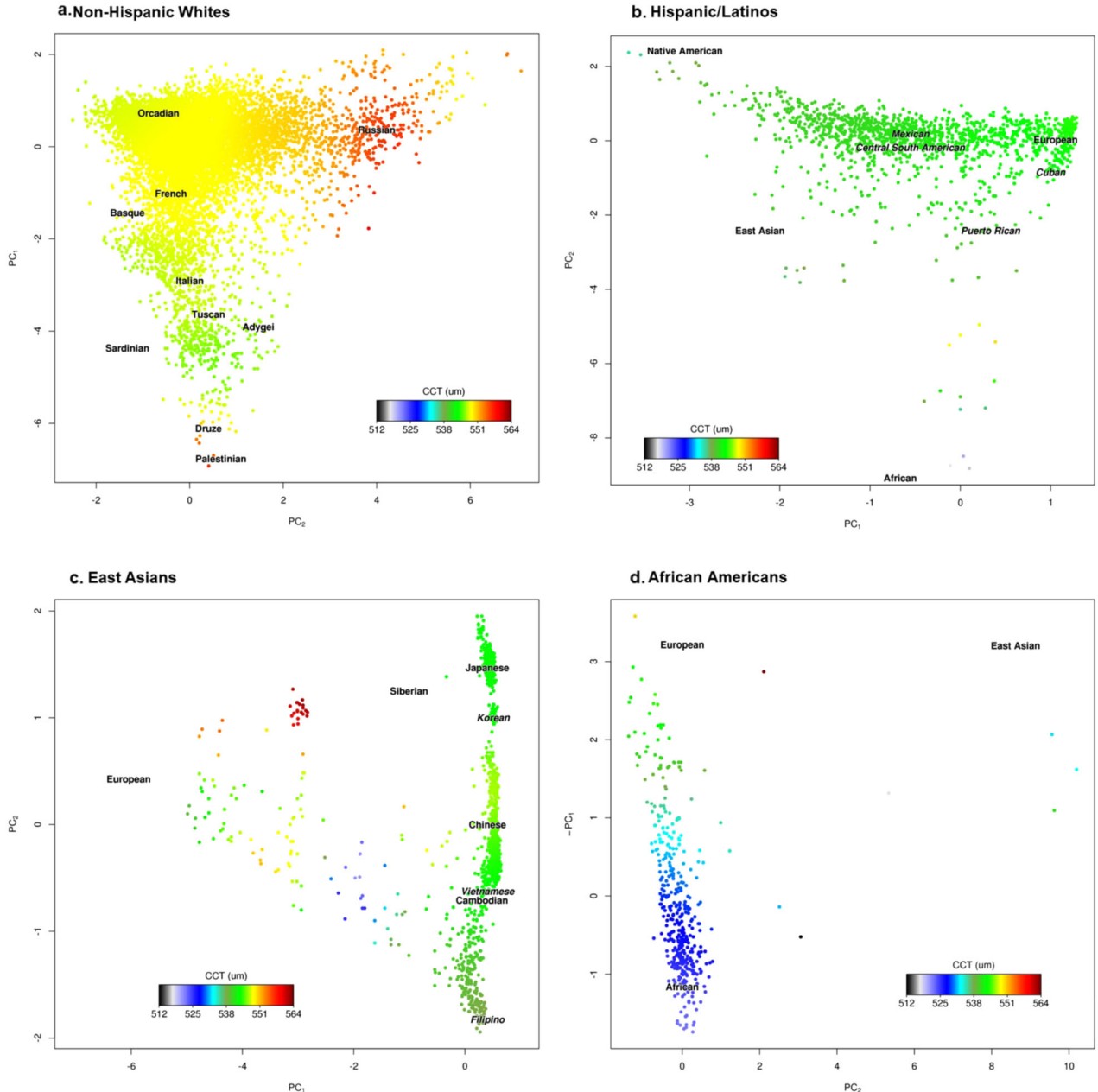

**Fig. 1 Plots of CCT distribution versus genetic ancestry in GERA.** CCT distribution is indicated on a color scale, standardized across groups, with warmer colors indicating thicker CCT. Axes reflect the first two principal components of ancestry in each group. The phenotype distribution was smoothed over the PCs (within the individuals in each respective figure), which were divided by their standard deviation for interpretability (see Methods). Human Genome Diversity Project populations are plotted at their relative positions in each figure. Human Genome Diversity Project populations are presented in a plain font, and GERA populations are presented in italics font. **a** non-Hispanic whites, **b** Hispanic/Latinos, **c** East Asians, and **d** African Americans.

(GERA + IGGC Asians only) did not result in the identification of additional novel genome-wide significant findings (Supplementary Fig. 7).

**Conditional analysis identified additional loci.** Conditional and joint (CoJo) analysis in the combined (GERA + IGGC) meta-analysis (full description in Methods) revealed 24 additional independent SNPs within the identified genomic regions, including the SNP rs72755233 at *ADAMTS17* on chromosome 15 which was recently identified in a GWAS of CCT conducted in the Icelandic deCODE health study[8]. Among those 24 independent SNPs, 10 have not been previously associated with CCT and

are not proxy variants of previously reported SNPs (Supplementary Data 5), resulting in a total of 98 independent genome-wide significant signals. After Bonferroni correction (0.05/98 SNPs tested), we found that the betas were not significantly different between GERA non-Hispanic whites and GERA Hispanic/Latinos (Supplementary Data 6).

**Array heritability estimate for CCT and variance explained.** We then estimated SNP-based heritability in the GERA non-Hispanic white ethnic group (the largest group of individuals from GERA) using GCTA[16], and we found an SNP-based heritability estimate of 42.5% (SE = 3.3%). When included together as a genetic risk

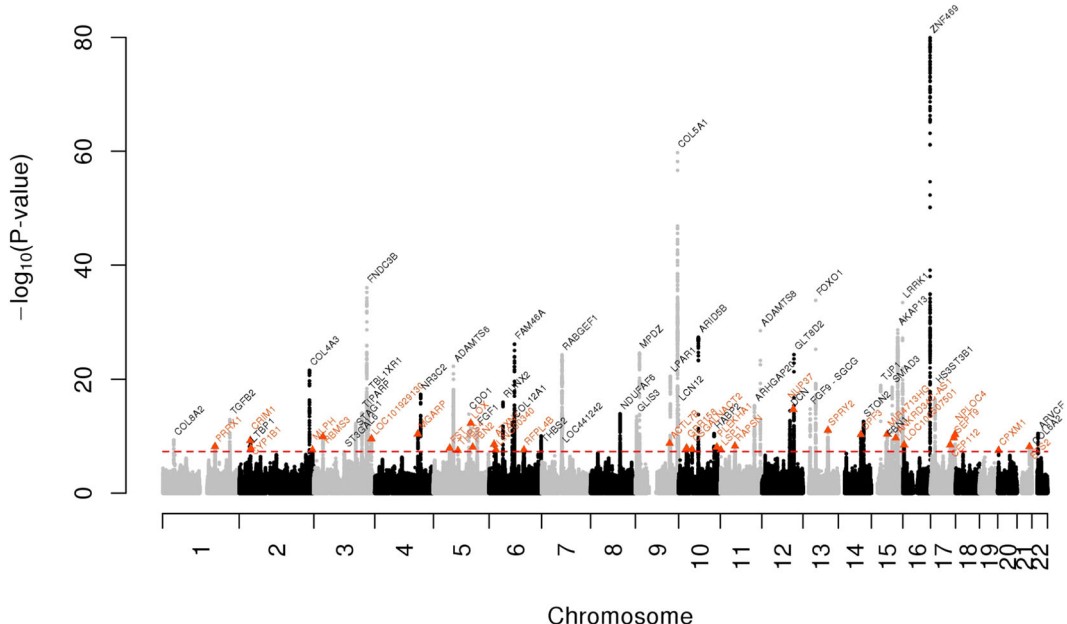

**Fig. 2 Manhattan plot of the GWAS meta-analysis (GERA+IGGC) for CCT.** Association results (–log10 P-values) are plotted for each chromosome. Names of loci and lead variants are indicated in color: previously identified loci are in black, and orange triangles indicate lead variants at novel loci.

**Table 2 Novel CCT loci identified in the combined (GERA+IGGC) GWAS meta-analysis.**

| SNV | Chr | Pos | Locus | Alleles | Combined meta-analysis | | | | | GERA | | IGGC | |
|---|---|---|---|---|---|---|---|---|---|---|---|---|---|
| | | | | | β (SE) | P | Q | I | | β (SE) | P | β (SE) | P |
| rs10800558 | 1 | 170828021 | PRRX1 | C/A | 1.38 (0.24) | $6.6 \times 10^{-9}$ | 0.30 | 6.67 | | 1.09 (0.37) | 0.0029 | 1.59 (0.31) | $3.7 \times 10^{-7}$ |
| rs848546 | 2 | 36704767 | CRIM1 | A/G | −1.55 (0.25) | $5.4 \times 10^{-10}$ | 0.37 | 0 | | −1.81 (0.39) | $2.9 \times 10^{-6}$ | −1.36 (0.33) | $3.0 \times 10^{-5}$ |
| rs1800440 | 2 | 38298139 | CYP1B1 | T/C | −1.89 (0.33) | $1.4 \times 10^{-8}$ | 0.88 | 0 | | −1.83 (0.50) | 0.00026 | −1.93 (0.44) | $1.4 \times 10^{-5}$ |
| rs880930 | 2 | 238396872 | MLPH | G/C | 1.58 (0.29) | $2.7 \times 10^{-8}$ | 0.34 | 0 | | 1.26 (0.44) | 0.0040 | 1.82 (0.37) | $1.2 \times 10^{-6}$ |
| rs11917483 | 3 | 29393868 | RBMS3 | T/C | −1.66 (0.26) | $1.2 \times 10^{-10}$ | 0.92 | 0 | | −1.63 (0.39) | $2.5 \times 10^{-5}$ | −1.68 (0.35) | $1.1 \times 10^{-6}$ |
| rs62292788 | 3 | 187016877 | LOC101929130 | G/A | 1.56 (0.25) | $2.9 \times 10^{-10}$ | 0.17 | 48.13 | | 1.97 (0.38) | $3.2 \times 10^{-7}$ | 1.27 (0.32) | $8.1 \times 10^{-5}$ |
| rs2320163 | 4 | 140174641 | MGARP | G/C | 1.59 (0.24) | $3.7 \times 10^{-11}$ | 0.94 | 0 | | 1.57 (0.37) | $2.0 \times 10^{-5}$ | 1.61 (0.32) | $4.3 \times 10^{-7}$ |
| rs7737693 | 5 | 52602941 | FST | C/T | 1.40 (0.25) | $1.3 \times 10^{-8}$ | 0.49 | 0 | | 1.21 (0.37) | 0.0011 | 1.54 (0.33) | $2.4 \times 10^{-6}$ |
| rs35351529 | 5 | 79390222 | THBS4 | T/C | 2.43 (0.44) | $3.2 \times 10^{-8}$ | 0.18 | 43.25 | | 3.05 (0.64) | $2.1 \times 10^{-6}$ | 1.88 (0.60) | 0.0017 |
| rs2731646 | 5 | 121420919 | LOX | A/G | 1.73 (0.24) | $5.2 \times 10^{-13}$ | 0.92 | 0 | | 1.76 (0.37) | $1.5 \times 10^{-6}$ | 1.71 (0.32) | $7.2 \times 10^{-8}$ |
| rs154001 | 5 | 127685135 | FBN2 | C/T | 1.54 (0.24) | $8.8 \times 10^{-15}$ | 0.067 | 70.21 | | 2.08 (0.40) | $1.9 \times 10^{-7}$ | 1.10 (0.36) | 0.0022 |
| rs6459472 | 6 | 16522607 | ATXN1 | A/G | 1.45 (0.24) | $2.6 \times 10^{-9}$ | 0.17 | 47.74 | | 1.06 (0.37) | 0.0040 | 1.74 (0.32) | $6.7 \times 10^{-8}$ |
| rs9350413 | 6 | 22067279 | LINC00340 | T/A | −1.42 (0.24) | $2.5 \times 10^{-8}$ | 0.76 | 0 | | −1.51 (0.38) | $6.9 \times 10^{-5}$ | −1.35 (0.35) | $9.3 \times 10^{-5}$ |
| rs7749695 | 6 | 113376787 | RFPL4B | T/C | 1.48 (0.27) | $2.7 \times 10^{-8}$ | 0.40 | 0 | | 1.76 (0.42) | $2.8 \times 10^{-5}$ | 1.30 (0.35) | 0.00017 |
| rs10124621 | 9 | 111488128 | ACTL7B | G/T | −1.61 (0.27) | $1.9 \times 10^{-9}$ | 0.54 | 0 | | −1.79 (0.40) | $7.5 \times 10^{-6}$ | −1.46 (0.36) | $5.1 \times 10^{-5}$ |
| rs4623781 | 10 | 25803024 | GPR158 | A/T | 1.35 (0.25) | $1.3 \times 10^{-8}$ | 0.45 | 0 | | 1.13 (0.37) | 0.0021 | 1.50 (0.31) | $1.2 \times 10^{-6}$ |
| rs2505507 | 10 | 43644824 | CSGALNACT2 | C/T | 1.53 (0.27) | $2.1 \times 10^{-8}$ | 0.27 | 18.15 | | 1.87 (0.41) | $5.7 \times 10^{-6}$ | 1.26 (0.36) | 0.00051 |
| rs4311997 | 10 | 124179299 | PLEKHA1 | C/T | 1.35 (0.23) | $8.3 \times 10^{-9}$ | 0.85 | 0 | | 1.40 (0.36) | 0.00011 | 1.31 (0.31) | $1.9 \times 10^{-5}$ |
| rs688601 | 11 | 1891284 | LSP1 | T/A | −1.67 (0.30) | $2.7 \times 10^{-8}$ | 0.55 | 0 | | −1.92 (0.51) | 0.00019 | −1.54 (0.37) | $3.3 \times 10^{-5}$ |
| rs3740685 | 11 | 47468791 | RAPSN | C/T | 1.50 (0.26) | $5.7 \times 10^{-9}$ | 0.77 | 0 | | 1.59 (0.40) | $6.0 \times 10^{-5}$ | 1.44 (0.34) | $2.3 \times 10^{-5}$ |
| rs4611262 | 12 | 102502759 | NUP37 | T/C | 2.37 (0.30) | $2.0 \times 10^{-15}$ | 0.21 | 35.97 | | 2.80 (0.45) | $6.2 \times 10^{-10}$ | 2.05 (0.40) | $2.7 \times 10^{-7}$ |
| rs1176321 | 13 | 81227243 | SPRY2 | G/A | 1.77 (0.26) | $9.3 \times 10^{-12}$ | 0.85 | 0 | | 1.83 (0.39) | $2.9 \times 10^{-6}$ | 1.73 (0.35) | $7.0 \times 10^{-7}$ |
| rs28667150 | 14 | 73137234 | DPF3 | G/A | −1.64 (0.25) | $5.3 \times 10^{-11}$ | 0.52 | 0 | | −1.83 (0.38) | $1.5 \times 10^{-6}$ | −1.50 (0.33) | $6.6 \times 10^{-6}$ |
| rs12898341 | 15 | 51456385 | MIR4713HG | C/T | −1.59 (0.24) | $4.2 \times 10^{-11}$ | 0.42 | 0 | | −1.82 (0.37) | $7.7 \times 10^{-7}$ | −1.43 (0.32) | $9.0 \times 10^{-6}$ |
| rs12324079 | 15 | 79501867 | ANKRD34C−AS1 | T/G | −1.54 (0.24) | $2.0 \times 10^{-10}$ | 0.65 | 0 | | −1.67 (0.37) | $7.0 \times 10^{-6}$ | −1.44 (0.32) | $6.2 \times 10^{-6}$ |
| rs4785955 | 16 | 4297151 | LOC100507501 | G/T | 1.79 (0.30) | $3.3 \times 10^{-9}$ | 0.50 | 0 | | 1.57 (0.45) | 0.00049 | 1.98 (0.41) | $1.4 \times 10^{-6}$ |
| rs7211723 | 17 | 63854425 | CEP112 | A/C | −1.62 (0.27) | $3.4 \times 10^{-9}$ | 0.63 | 0 | | −1.77 (0.42) | $2.7 \times 10^{-5}$ | −1.50 (0.36) | $2.8 \times 10^{-5}$ |
| rs448203 | 17 | 75495065 | SEPT9 | T/C | 1.84 (0.29) | $2.1 \times 10^{-10}$ | 0.54 | 0 | | 2.01 (0.41) | $7.8 \times 10^{-7}$ | 1.66 (0.41) | $5.3 \times 10^{-5}$ |
| rs11650127 | 17 | 79572253 | NPLOC4 | G/A | −1.96 (0.30) | $4.8 \times 10^{-11}$ | 0.49 | 0 | | −2.15 (0.40) | $1.1 \times 10^{-7}$ | −1.73 (0.44) | $8.1 \times 10^{-5}$ |
| rs13036662 | 20 | 2784053 | CPXM1 | G/A | −2.11 (0.38) | $3.2 \times 10^{-8}$ | 0.73 | 0 | | −1.95 (0.60) | 0.0012 | −2.22 (0.49) | $6.8 \times 10^{-6}$ |
| 21:40195541 | 21 | 40195541 | ETS2 | G/GT | 1.46 (0.25) | $7.2 \times 10^{-9}$ | 0.27 | 18.75 | | 1.78 (0.38) | $3.0 \times 10^{-6}$ | 1.22 (0.34) | 0.00033 |

score in a linear-regression model, the 98 lead SNPs identified in the current study (74 from the combined meta-analysis + 24 from the COJO analysis) collectively explained up to 14.2% of the CCT variance in GERA non-Hispanic whites (Supplementary Data 7). To determine whether the 98 CCT-loci explain the observed association of genetic ancestry with CCT variation within GERA African Americans, we repeated the ancestry analysis, including a genetic risk score. After accounting for the effect of the 98 identified SNPs in the genetic ancestry model, the African (versus European) ancestry (PC1) association was no longer significant ($P = 0.12$) within the GERA African Americans. This suggests that the identified CCT-loci explain most of the ancestry effects in African Americans.

**SNP prioritization and annotations**. To prioritize variants within the 74 genomic regions identified in the combined (GERA + IGGC) meta-analysis, we computed each variant's ability to explain the observed signal and derived the smallest set of variants that included the causal variant with 95% probability[17]. In each of the 74 autosomal loci, the corresponding 74 credible sets contained from 1 to 143 variants (1554 total variants, Supplementary Data 8). Interestingly, among the 31 sets representing newly identified CCT genomic regions, three sets included a unique variant (i.e. CYP1B1 rs1800440, FBN2 rs154001, and NUP37 rs4611262 with 99.1%, 97.9%, and 100% posterior probability of being the causal variants, respectively), suggesting that those variants may be the true causal variants.

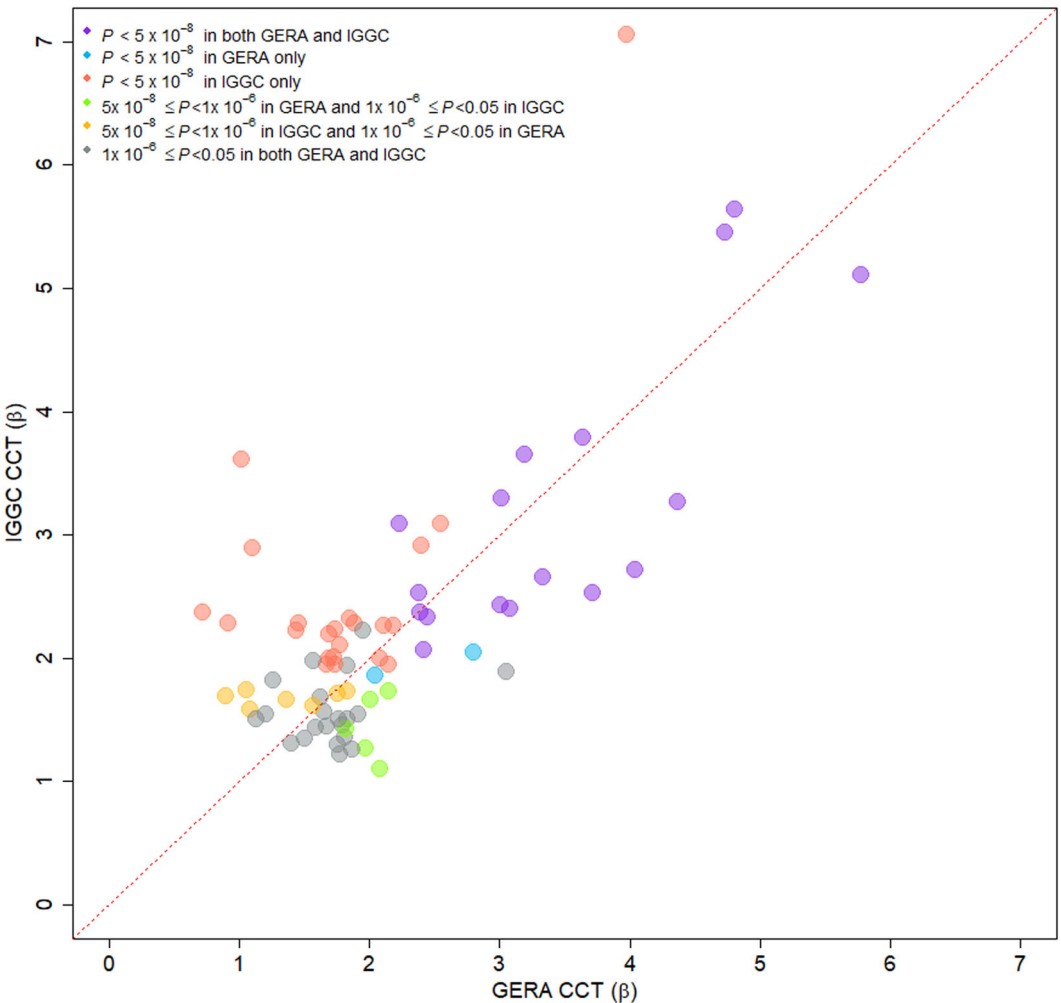

**Fig. 3 Correlation of effect sizes for CCT between GERA and IGGC cohorts for the lead 74 CCT-associated SNPs.** The 74 CCT-associated SNPs (novel and previously reported) were identified in the combined (GERA+IGGC) meta-analysis ($N = 44{,}039$ individuals). Comparison of regression coefficients in IGGC (y axis) and GERA (x axis; correlation coefficient = 0.72).The 74 CCT SNPs are displayed in different colors according to their association P-values in IGGC and in GERA.

**Gene prioritization**. To prioritize genes within the 74 GERA + IGGC genomic regions, we used the DEPICT[18] integrative tool. DEPICT gene prioritization analysis detected 14 genes, of which 4 were within novel CCT-associated loci, to prioritize after false-discovery rate (FDR) correction (Supplementary Data 9). These included: *C1orf129* (near *PRRX1* and also known as *MROH9*) on chromosome 1, *CYP1B1* on chromosome 2, *LOX* on chromosome 5, and *LSP1* on chromosome 11.

**Gene expression in human ocular tissues**. We then evaluated the ocular expression levels of genes at CCT loci that contained associated 95% credible set variants across various human eye tissues (i.e. choroid retinal pigment epithelium, ciliary body, cornea, iris, lens, optic nerve, optic nerve head, retina, sclera, and trabecular meshwork). Among the genes at novel CCT loci, *PLEKHA1*, which encodes the pleckstrin homology domain containing A1, was highly expressed (PLIER number>200) in the cornea, lens, and the trabecular meshwork according to the Ocular Tissue Database (OTDB)[19] (Supplementary Data 10). Similarly, *CRIM1*, which encodes the cysteine rich transmembrane BMP regulator 1, was highly expressed in the cornea and the lens.

**Biological pathway annotations and prioritization**. DEPICT tissue-enrichment analysis highlighted 17 significantly associated (FDR < 0.05) tissues or cell type annotations, consistent with recent findings[14]; 4 annotations pertained to fibroblast and collagen-rich tissues such as cartilage, joints, synovial membrane, and joint capsule (Supplementary Data 11). An additional 5 annotations involved granulation tissue, cicatrix, keloid, chorion, and extraembryonic membranes, and 5 annotations included cell types such as, osteoblasts, mesenchymal stem cells, chondrocytes, stromal cells, and fibroblasts. DEPICT gene-set enrichment analysis did not detect pathway to prioritize after FDR correction; however, nominal evidence was found for previously reported gene-sets[14], including those involved in the regulation of epithelial to mesenchymal transition, extracellular matrix (ECM) organization, and collagen formation (Supplementary Data 12).

For comparison with previous works[14,20], we also conducted a pathway analysis using VEGAS software[21] to assess enrichment in 9732 pathways or gene-sets derived from the Biosystem's database. Using a 10-kb window in the VEGAS2 computation, we found that 25 pathways/gene-sets were significantly enriched after correcting for multiple testing ($P < 5.14 \times 10^{-6}$) (Supplementary Data 13), compared to 23 pathways/gene-sets previously identified[14]. Similarly, most of these pathways/gene-sets

contribute to the function of the extracellular matrix and collagen. In addition, we identified gene-sets related to head/face morphogenesis and development.

**Effect of the 98 CCT-associated loci on keratoconus.** As previous studies have reported that CCT is significantly lower in eyes with keratoconus compared to normal eyes[1,22], we evaluated the effect estimates of the 98 CCT-SNPs identified in the current study (74 from the combined meta-analysis + 24 from the COJO analysis) between CCT and keratoconus in GERA. Our GERA keratoconus cohort consisted of 207 cases and 97,375 controls (Table 1). We confirmed the negative correlation of effect estimates between CCT and keratoconus ($R^2 = -0.32$, $P = 1.30 \times 10^{-3}$; Supplementary Fig. 8), as previously reported[14]. We then evaluated whether the 98 CCT-SNPs identified in the current study were also associated with keratoconus in a multiethnic meta-analysis combining the GERA keratoconus cohort, and an independent cohort from the UK of 2353 patients with keratoconus[23] (see full description in the Methods). Among the 97 CCT-SNPs available, 20 were significantly associated with keratoconus after correction for multiple testing ($P < 5.15 \times 10^{-4}$, 0.05/97), including 12 at genome-wide level of significance (Supplementary Data 14). These included SNPs at *ZNF469*, *FOXO1*, *MPDZ*, *SMAD3*, and *COL5A1*, consistent with previous findings[14] and with the expected direction of effect; but also SNPs at novel CCT-loci: *LOX* and *FST*. Further, an additional 18 were associated with keratoconus at a nominal level ($P < 0.05$).

**Effect of the 98 CCT-associated loci on POAG.** As observational epidemiologic studies have reported that thinner CCT is associated with an increased risk of glaucoma[1,3,4], we evaluated the effect estimates of the 98 CCT-SNPs identified in the current study between CCT and POAG in GERA. Our GERA POAG cohort consisted of 4986 cases and 58,426 controls (Table 1), and consistent with previous findings[14], no correlation in effect estimates was found between CCT and POAG ($R^2 = -0.16$, $P = 0.12$; Supplementary Fig. 9). We then investigated whether the 98 CCT-SNPs identified in the current study were also associated with POAG in GERA. Associations with glaucoma (subtype unspecified) were then confirmed in UKB[24]. Among the 98 lead CCT-associated SNPs identified in the current study, only SNP rs3740685 in *RAPSN* was associated with POAG in GERA after multiple testing correction ($P = 1.9 \times 10^{-4}$; Supplementary Data 15). Consistently, *RAPSN* rs3740685 was nominally associated with glaucoma (subtype unspecified) in UKB ($P = 0.0017$). Visual inspection of the association signals in the *RAPSN* region based on GERA results revealed that the lead SNPs for POAG (rs2167079) and CCT (rs3740685) are different (Supplementary Fig. 10). Further, those lead SNPs are relatively distant from one another (198.5 kb apart) and are only moderately correlated in European-ancestry populations ($r^2 = 0.40$; D′ = 0.63), suggesting that they may represent different signals. When we repeated the POAG association analysis in GERA, conditioning on our most strongly associated CCT SNP (rs3740685) from our GERA meta-analysis, *RAPSN* rs2167079 remained significant ($P = 0.0098$), suggesting that our lead associated SNPs for CCT and POAG represent different signals at the *RAPSN* locus. This locus has previously demonstrated genome-wide significant association with intraocular pressure[25–27], which is an important risk factor for developing POAG.

**Two-sample Mendelian randomization.** In GERA, POAG was significantly associated with lower CCT after adjusting for age, sex, ethnic group, and CCT measurement type ($\beta = -12.57$ and $P = 2.33 \times 10^{-66}$; Supplementary Data 16). This significant

association between POAG and lower CCT was true for all the GERA ethnic groups. Further, consistent with previous studies[28–30], we observed that GERA POAG patients have, on average, thinner CCTs than controls (Mean ± SD: 537.0 ± 34.7 for POAG cases vs. 549.5 ± 34.4 for controls; $P = 3.29 \times 10^{-74}$). To clarify whether the observational association between CCT and POAG risk is consistent with a causal relationship, we conducted a two-sample Mendelian randomization analysis[31]. We built MR models using as instruments genetic effects previously SNP alleles associated with CCT in the IGGC European-specific meta-analysis[14] (exposure) and associations with POAG[32] (outcome of interest) observed in the GERA non-Hispanic whites (full description in Methods). The MR Egger intercept test suggested no horizontal pleiotropy (intercept = 0.027; SE = 0.020; $P = 0.19$) and we observed no notable heterogeneity across instrument SNP effects ($Q = 32.81$; $P = 0.12$). Our two-sample Mendelian randomization analysis did not detect any evidence of causal relationship between CCT and POAG (inverse variance–weighted (IVW) method: OR = 1.00, se = 1.00, $P = 0.088$), and the weighted median and MR Egger analyses yielded similar results (Supplementary Data 17 and Supplementary Figs. 11–13).

**Discussion**
Our analysis of 44,039 individuals identified 41 novel loci significantly associated with CCT inter-individual variation. These loci, along with those previously reported, account for 14.2% of phenotypic variation and largely explain the genetic ancestry association of thinner CCT observed in African Americans. We found that 20.6% (20/97) of the CCT-loci were significantly associated with keratoconus but only the *RAPSN* locus was associated with POAG after correction for multiple testing. It appears that the associations at the *RAPSN* locus may represent two independent signals. Finally, our MR analyses suggest, for the first time to our knowledge, that thinner CCT might not causally increase the risk of POAG.

Although GWAS-identified associations do not directly highlight a specific gene, our study revealed potential candidate genes, including *LOX, FBN2, SPRY2*, and *CRIM1*, which have all been linked to cornea and eye development. *LOX* encodes a member of the lysyl oxidase family of proteins, which is responsible for the cross-linking of collagens and elastin[33]. *LOX* is differentially expressed in keratoconus epithelium[34], and *LOX* variants lead to increased susceptibility to keratoconus[35,36], and are associated with intraocular pressure variation[25–27]. *FBN2* encode the fibrillin 2 which has a crucial role in ocular morphogenesis in mice[37] and is expressed in the corneal stroma but not in the corneal epithelium of mice heterozygous for the micropinna microphthalmia (Mp) mutation, a 660-kb inversion on chromosome 18 that disrupts the *Fbn2* gene[38,39]. Fbn2 is involved in the corneal epithelial homeostasis of Mp/+ mice[39], and rare and common variants in *FBN2* have been shown to be associated with macular degeneration in human[40,41]. Our CCT study also implicated *SPRY2* which encodes a protein belonging to the sprouty family. This *SPRY2* gene is involved in regulating corneal epithelial cell proliferation and differentiation, enabling eyelid closure[42,43]. Our study also identified *CRIM1*, which encodes a transmembrane protein containing six cysteine-rich repeat domains and an insulin-like growth factor-binding domain. Importantly, *CRIM1* is involved in eye development in human and mouse[44,45] and in the corneal response to ultraviolet and pterygium development[46], and is required for maintenance of the ocular lens epithelium[47].

Similarly, our study revealed potential biological pathways and relevant tissues involved in CCT variation that are pertained to the function of fibroblast and collagen-rich tissues, as well as the regulation of epithelial to mesenchymal transition, and the

organization of the extracellular matrix, consistent with previous works[14]. Functional follow-up experiments in cell lines or animal models may confirm the involvement of these genes and biological pathways in CCT variation and reveal the underlying mechanisms of CCT-related vision disorders.

Many of the CCT-associated loci identified in this study, are also associated with other eye conditions, particularly *NPLOC4*, *CYP1B1*, *RBMS3*, and *PLEKHA1*. *NPLOC4* encodes the NPL4 homolog, ubiquitin recognition factor, and polymorphisms at this locus have been previously reported to be associated with myopia, age-related macular degeneration, eye color, and recently with corneal or refractive astigmatisms, strabismus and macular thickness[48–53]. Mutations in *CYP1B1*, which encodes a member of the cytochrome P450 superfamily of enzymes, involved in eye development[54], are associated with primary congenital glaucoma[55,56]. Similarly, in the current CCT study we identified *RBMS3* which encodes the RNA binding motif single stranded interacting protein 3, and a recent GWAS reported a strong association between *RBMS3* locus and increased risk of exfoliation syndrome[57]. Our study also identified as a CCT locus *PLE-KHA1*, which encodes a pleckstrin homology domain-containing adapter protein. Polymorphisms in *PLEKHA1* are associated with age-related macular degeneration[58,59].

We recognize several potential limitations of our study. First, the 'non-cases' of the keratoconus GERA study may include some potential cases with other ophthalmic conditions, which may result in underestimates of the effects of individual CCT-associated SNPs if those conditions are also associated with the risk of keratoconus. Second, glaucoma diagnoses in UKB were based on self-reported data, and the subtypes of glaucoma were unspecified, which may result in underestimates of the effects of individual SNPs due to phenotype misclassification. However, our glaucoma results were consistent across GERA and UKB. Third, to date, it has been difficult to discern whether associations between CCT and POAG are truly causal or biased due to confounding associated with traditional observational studies[2–4]. Our MR analysis failed to detect alteration in CCT as a causal risk factor for POAG. This result is unexpected, and it is possible that other factors (e.g. environmental, epigenetics) that have not been taken into consideration in the current study, might influence the relationship between CCT and POAG. Future studies will be needed to clarify further the relationship between CCT and POAG.

In summary, our study identified 98 independent loci associated with CCT, of which 41 were novel. In addition to doubling the number of CCT-associated loci reported, this study explains up to 14.2% of the variance of CCT. The loci also explain variation among African Americans due to genetic ancestry. Our Mendelian randomization analysis did not support the idea that a thinner CCT causally increase the risk of POAG. Altogether, this large study of CCT increases substantially our understanding of the etiology of CCT variation and may open new avenues of investigation into human ocular traits and their relationship to the risk of vision disorders.

## Methods

**GERA cohort.** The Genetic Epidemiology Research in Adult Health and Aging (GERA) cohort consists of 110,266 adults, 18 years and older, who are consented participants in the Research Program on Genes, Environment, and Health (RPGEH). The GERA participants are members of the Kaiser Permanente Northern California (KPNC) integrated health care delivery system, and most have ongoing longitudinal records from vision examinations. For the current study, 18,129 GERA participants from four ethnic groups who had at least one recorded CCT measurement on both eyes during the same visit between June 2014 and January 2018 were included (Table 1). All study procedures were approved by the Institutional Review Board of the Kaiser Permanente Northern California Institutional Review Board. Written informed consent was obtained from all participants.

**CCT measurement.** CCT was measured in GERA using the DGH-550 or DGH-55 ultrasonic (contact) pachymeter (DGH Technology Inc.; Exton, PA)[4], or a non-contact optical biometer (Lenstar LS900, Haag-Streit, Köniz, Switzerland), and recorded for both eyes in the electronic health records. Patients with single eye measurements were removed. We also excluded 1106 patients who had ocular conditions which may influence CCT, including patients with Fuchs dystrophy, keratoconus, history of corneal refractive surgery, corneal transplantation, or laser vision surgery. For patients with both types of measurement (i.e., ultrasonic pachymeter and non-contact biometer) available, we selected the non-contact biometer measurements. Because the means of the distributions of the two types of measurements slightly differed, we transformed all CCT values to the standard normal distribution scale, with $\mu = 0$ and $\sigma = 1$, i.e. N(0,1). Then, the mean standardized CCT of both eyes and standard deviation (sd) were assessed for each patient. Outliers ($N = 9$) defined by large left-right differences (i.e., beyond 4 sd of the overall standardized distribution of left-right differences) were removed (Supplementary Fig. 14). Finally, for consistency of CCT scale between GERA and IGGC, we rescaled standardized CCT values, as follows:

$$\text{CCT}_{\text{Final}} = \text{CCT}_{\text{Standardized value}} * \text{sd}(\text{CCT}_{\text{non-contact}}) + \text{mean}(\text{CCT}_{\text{non-contact}})$$

**Genotyping and imputation.** GERA DNA samples were genotyped at the Genomics Core Facility of the University of California, San Francisco (UCSF) on four ethnicity-specific Affymetrix Axiom arrays (Affymetrix, Santa Clara, CA, USA) optimized for individuals of European, Latino, East Asian, and African American ancestry[60,61]. Genotype QC (quality control) procedures were performed on an array-wise basis[62], as follows: SNPs with initial genotyping call rate ≥97%, allele frequency difference (≤0.15) between males and females for autosomal markers, and genotype concordance rate (>0.75) across duplicate samples were included. About 94% of samples and more than 98% of genetic markers assayed passed QC procedures. In addition to those QC criteria, SNPs with a minor allele frequency <1% were removed.

Imputation was also conducted on an array-wise basis. After the pre-phasing of genotypes with Shape-IT v2.r2717958[63], variants were imputed from the cosmopolitan 1000 Genomes Project reference panel (phase I integrated release; http://1000genomes.org) using IMPUTE2 v2.3.059[64]. As a QC metric, we used the info $r^2$ from IMPUTE2, which is an estimate of the correlation of the imputed genotype to the true genotype[65]. We excluded variants with an imputation $r^2 < 0.3$, and restricted to SNPs that had a minor allele count ≥20.

**Principal components analysis.** We used Eigenstrat[66] v4.2 to calculate the principal components (PCs) on each of the four GERA ethnic groups[15]. The top 10 ancestry PCs were included as covariates for the non-Hispanic whites, while the top six ancestry PCs were included for the three other ethnic groups. The percentage of Ashkenazi (ASHK) ancestry was also used as a covariate for the non-Hispanic whites to adjust for genetic ancestry[15]. Association of each PCs and ASHK ancestry with CCT are reported in Supplementary Data 1.

**Genetic ancestry analysis.** A full description of the ancestry analyses in GERA is provided in Banda et al.[15]. The CCT distribution by the ancestry PCs for each GERA ethnic groups is illustrated on Fig. 1. To create these plots, a smoothed distribution of each individual $i$'s CCT phenotype was created using a radial kernel density estimate weighted on the distance to each other $j$th individual, as follows: $\sum_j \phi\left(\left\{d(i,j)/\max_{i,j'}[d(i,j')]*k\right\}\right)$, where $\phi(.)$ is the standard normal density distribution, $k$ is the smooth value (5 for non-Hispanic whites; and 15 for East Asians, Hispanic/Latinos, and African-Americans), and $d(i', j')$ is the Euclidean distance based on the first two PCs (Fig. 1). For visual representation of different groups, we derived the ethnicity and/or nationality subgroup labels from GERA or the Human Genome Diversity Project[15].

**Association analysis in GERA.** Each of the four GERA ethnic groups (non-Hispanic whites, Hispanic/Latinos, East Asians, and African-Americans) were first analyzed individually. We performed a linear regression of CCT and each SNP using PLINK[67] v1.9 (www.cog-genomics.org/plink/1.9/) with the following covariates: age, sex, ancestry PCs, and CCT measurement type (i.e., ultrasonic pachymeter or non-contact optical biometer). Data from each genetic variant were modeled using additive dosages to account for the uncertainty of imputation[68]. We then performed a GERA meta-analysis of CCT to combine the results of the four ethnic groups using the package "meta" of R (https://www.R-project.org).

**International Glaucoma Genetics Consortium.** The IGGC study was a meta-analysis of 25,910 participants from 19 CCT cohorts of European (14 cohorts) and Asian descent (5 cohorts)[14]. A full description of the individual cohorts are provided in previous publications[69,70]. GWAS summary statistics from the study of Iglesias et al.[14] were publicly accessible at http://hdl.handle.net/10283/2976.

**Meta-analysis.** To combine the study results of Iglesias et al.[14] with our GERA meta-analysis, we conducted a fixed-effect meta-analysis. Heterogeneity index, $I^2$ (0–100%) and P-value for Cochrane's Q statistic among studies were assessed. For

each locus, the top genetic variant was defined as the most significant variant within a 2-Mb window, and novel loci were defined as those that were located over 1 Mb apart from any previously reported locus.

**Conditional and joint analysis**. To potentially identify independent signals within the 74 identified genomic regions, we performed a multi-SNP-based conditional and joint association analysis (COJO)[71], which is implemented in the Genome-wide Complex Trait Analysis (GCTA) integrative tool[72]. This COJO analysis was conducted on the combined (GERA+IGGC) meta-analysis results. We first conducted the COJO analysis on each of the individual ethnic groups (European-specific samples (GERA and IGGC Europeans), GERA Hispanic/Latinos, Asian-specific samples (GERA and IGGC Asians), and GERA African Americans). To calculate linkage disequilibrium (LD) patterns we used the following reference panels: 10,000 random samples from GERA non-Hispanic white ethnic group as a reference panel for European samples, 8565 samples from GERA Hispanic/Latino ethnic group for Hispanic/Latino samples, 7518 samples from GERA East Asian ethnic group for Asian samples, and 3161 samples from GERA African American ethnic group for African American samples. We then meta-analyzed the four ethnic groups COJO results by calculating fixed effects summary estimates for combining the P-values. For this COJO analysis we considered a P-value $<5 \times 10^{-8}$ as the significance threshold.

**SNP-based heritability and variance explained**. GWAS heritability estimate was obtained for CCT in GERA non-Hispanic whites (the largest ethnic group of GERA) using the GCTA software[16,72]. To estimate the proportion of variance in CCT explained by the 98 identified CCT-associated SNPs, we performed a REML (restricted maximum likelihood) analysis (GREML) using GCTA[73].

**Variants prioritization**. To prioritize genetic variants, we used a Bayesian approach (CAVIARBF)[17]. For each of the 74 associated signals identified in the combined (GERA+IGGC) meta-analysis, we computed each variant's capacity to explain the identified signal within a 2-Mb window (±1.0 Mb with respect to the original top variant) and derived the smallest set of variants that included the causal variant with 95% probability (95% credible set). A total of 1554 variants within 72 annotated genes were included in these 74 credible sets (Supplementary Data 8). For this CAVIARBF analysis, we used 10,000 random samples from GERA non-Hispanic white ethnic group as a reference panel to calculate LD patterns.

**Genes and biological pathways prioritization**. To prioritize genes and biological pathways, and highlight gene-set and tissue/cell enrichments within the 74 CCT genomic regions identified in the combined (GERA + IGGC) meta-analysis, we used the following integrative tool: DEPICT[18]. All independent genome-wide significant genetic variants in the combined (GERA + IGGC) meta-analysis served as input, and as the reference panel, we used 10,000 random samples from GERA non-Hispanic white ethnic group. Genes, gene-sets, tissue/cell annotations that achieved a nominal significance level of 0.05 after false-discovery rate (FDR) correction were subsequently prioritized.

To compare our pathways analysis results with recent findings[14,20], we also conducted a pathways analysis using the Versatile Gene-based Association Study - 2 version 2 (VEGAS2v02) web platform[21]. We first performed a gene-based association analysis on the combined (GERA + IGGC) meta-analysis results using the default '-top 100' test that uses all (100%) variants assigned to a gene to compute gene-based P-value. Gene-based analyses were conducted on each of the individual ethnic groups (European-specific samples (GERA and IGGC Europeans), GERA Hispanic/Latinos, Asian-specific samples (GERA and IGGC Asians), and GERA African Americans) using the appropriate reference panel: 1000 Genomes phase 3 European population, 1000 Genomes phase 3 American population, 1000 Genomes phase 3 Asian population, and 1000 Genomes phase 3 African population, respectively. Second, we performed pathways analyses based on VEGAS2 gene-based P-values. We tested enrichment of the genes defined by VEGAS2 in 9732 pathways or gene-sets (with 17,701 unique genes) derived from the Biosystem's database (https://vegas2.qimrberghofer.edu.au/biosystems20160324.vegas2pathSYM). We adopted the resampling approach to perform pathway analyses using VEGAS2 derived gene-based P-values considering the default '−10 kbloc' parameter as previously decribed[14]. We then meta-analyzed the four ethnic groups gene-based results using Fisher's method for combining the P-values.

**Ocular gene expression**. Expression of the genes ($N = 74$) that contained associated 95% credible set variants was evaluated in human ocular tissues using two publicly available databases: the OTDB[19] and EyeSAGE[74,75] publicly available at https://genome.uiowa.edu/otdb/ and http://neibank.nei.nih.gov/EyeSAGE/index.shtml, respectively. The OTDB consists of gene expression data for 10 eye tissues from 20 normal human donors, and the gene expression is described as Affymetrix Probe Logarithmic Intensity Error (PLIER) normalized value[19].

**Associations with keratoconus**. To identify keratoconus cases ($N = 207$) in GERA, we selected all participants who had at least one diagnosis of keratoconus by a Kaiser Permanente ophthalmologist based on the following ICD-9 diagnosis codes: 371.60, 371.61, and 371.62, and conducted a chart review of each of those cases (by Dr. Melles, a KPNC ophthalmologist). Our GERA keratoconus control group ($N = 97,375$) included all the non-cases.

To evaluate whether the 98 CCT-SNPs identified in the current study were also associated with keratoconus, we conducted a multiethnic meta-analysis combining the GERA keratoconus cohort, and an independent cohort of patients with keratoconus, recruited from specialist clinics at Moorfields Eye Hospital, London, United Kingdom; the recruitment methodology is the same as that described for a previously published subset of European ancestry[23]. Briefly, each participant was examined using tomography (Pentacam; Oculus), and the presence of keratoconus was confirmed using established criteria based on corneal thinning and corneal distortion[76]. A previous bilateral keratoplasty for keratoconus was also accepted as confirmation of disease status. Patients with syndromic disease and keratoconus (e.g., Down syndrome, Ehlers Danlos syndrome) were excluded. Controls were extracted from a pool of 80,000 randomly selected participants in the UK Biobank cohort. Exclusions included any individual with any ICD9 or ICD10 code for any corneal disease. The cases and controls were ethnically matched. In total, 2353 keratoconus cases from the Moorfields Eye Hospital and 37,360 controls from UK Biobank were included (1371 cases and 25,166 controls of European, 661 cases and 8009 controls of South Asian, and 321 cases and 4185 controls of African ancestry). Keratoconus cases and controls were genotyped using the Affymetrix UK Biobank Axiom Array.

**Associations with POAG in GERA**. Associations of CCT-associated SNPs with open-angle glaucoma (POAG) were also evaluated in the GERA cohort. POAG cases were diagnosed by a Kaiser Permanente ophthalmologist and were identified in the KPNC electronic health record system based on the International Classification of Diseases, Ninth Revision (ICD-9) diagnosis codes (i.e., ICD-9 codes 365.01, 365.1, 365.10, 365.11, 365.12, and 365.15) as previously reported[32]. In total, 4986 POAG cases were identified in GERA. Our POAG control group ($N = 58,426$) included all the non-cases, after excluding subjects who have one or more diagnosis of any type of glaucoma (ICD-9 code, 365.xx).

**Associations with glaucoma in UKB**. We then evaluated associations between CCT-associated genetic variants and glaucoma (subtype unspecified) in the multiethnic UK Biobank (UKB)[24,77]. In UKB, the glaucoma phenotype was assessed through a touchscreen self-report questionnaire completed at the Assessment Centre, via the question "Has a doctor told you that you have any of the following problems with your eyes?", and cases (N= 7,329) were defined as those reporting "glaucoma" (subtype unspecified). The control group included 169,561 individuals. For this confirmation analysis, as one phenotype and 89 genetic variants were tested (as only 89 out of the 98 CCT-SNPs were available in UKB), our P-value adjusted for Bonferroni correction was set as $P < 5.62 \times 10^{-4}$ (0.05/89).

**Mendelian randomization analyses for CCT and POAG**. To assess the causal relationships between CCT and POAG, we conducted two-sample Mendelian Randomization analyses using the TwoSampleMR package[31]. For CTT exposure risk factor, we used the following set of genetic instrument drawn on summary statistics data from the published Iglesias et al. European-specific meta-analysis[14]: lead SNPs previously reported as genome-wide significant ($P < 5 \times 10^{-8}$); after clumping SNPs for independence, 26 representative SNPs were retained. We built MR models using our GWAS summary associations for POAG[32] (outcome of interest) from GERA non-Hispanic whites. Here, we reported MR estimates using the inverse variance–weighted (IVW) method as MR estimates using weighted median and MR Egger methods yielded similar pattern of effects (Supplementary Data 17 and Supplementary Fig. 11). Further analyses were conducted, including horizontal pleiotropy, leave-one-SNP-out or single variant analyses (Supplementary Figs. 12 and 13).

**Reporting summary**. Further information on research design is available in the Nature Research Reporting Summary linked to this article.

## Data availability

The GERA genotype data are available upon application to the KP Research Bank (https://researchbank.kaiserpermanente.org/). The summary statistics generated in the study of Iglesias et al.[14] are available at http://hdl.handle.net/10283/2976. The combined (GERA+IGGC) meta-analysis GWAS summary statistics are available from the NHGRI-EBI GWAS Catalog (https://www.ebi.ac.uk/gwas/downloads/summary-statistics).

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

## Acknowledgements

We are grateful to the Kaiser Permanente Northern California members who have generously agreed to participate in the Kaiser Permanente Research Program on Genes, Environment, and Health. Support for participant enrollment, survey completion, and biospecimen collection for the RPGEH was provided by the Robert Wood Johnson Foundation, the Wayne and Gladys Valley Foundation, the Ellison Medical Foundation, and Kaiser Permanente Community Benefit Programs. Genotyping of the GERA cohort was funded by a grant from the National Institute on Aging, National Institute of Mental Health, and National Institute of Health Common Fund (RC2 AG036607 to C.S. and N.R.). Data analyses were facilitated by National Eye Institute (NEI) grant R01 EY027004 and by the National Institute of Diabetes and Digestive and Kidney Diseases (NIDDK) R01 DK116738 (E.J.). This work was also made possible in part by NIH-NEI EY002162—Core Grant for Vision Research, by the Research to Prevent Blindness Unrestricted Grant (UCSF, Ophthalmology). K.S.N. receives support from NEI grant EY022891, Research to Prevent Blindness William and Mary Greve Special Scholar Award, Marin Community Foundation-Kathlyn McPherson Masneri and Arno P. Masneri Fund, and That Man May See Inc. The funders had no role in study design, data collection and analysis, decision to publish, or preparation of the manuscript. The UK keratoconus study was funded by Moorfields Eye Charity and supported by infrastructure and funding from the National Institute for Health Research Biomedical Research Centre at Moorfields Eye Hospital and UCL Institute of Ophthalmology.

## Author contributions

H.C. and E.J. conceived and designed the study. T.J.H., M.N.K, N.R., C.S., and E.J. were involved in the genotyping and quality control. T.J.H. performed the imputation analyses. J.Y. and K.K.T., in collaboration with R.B.M., extracted phenotype data from EHRs. K.K.T., and J.Y. performed statistical analyses. Y.B. performed the ancestry principal components analyses. J.Y. performed in silico analyses. S.J.T and A.J.H clinically evaluated, sampled, and genotyped the Moorfields Eye Hospital keratoconus cohort. H.C., T.J.H., C.S., R.B.M., N.R., M.M.G, K.S.N., P.G.H., and E.J. interpreted the results of analyses. H.C., R.B.M., N.R., K.S.N., P.G.H., and E.J. contributed to the drafting and critical review of the manuscript.

## Competing interests

The authors declare no competing interests.
