## [Peer Review File · Communications Biology]

Reviewers' comments:

Reviewer #1 (Remarks to the Author):

Choquet et al. presented data on GWAS for central corneal thickness and identified 41 novel loci. The GWAS findings of this study are significant in the field to understand CCT genetics. The data presentation is comprehensive. However, some analysis aspects need to be clarified and in general, the discussion could cover more regarding underlying biological mechanisms for CCT.

Specific comments below:

- 1) For GREY cohorts, top variants explain 14.2% phenotypic variation in Non-Hispanic whites, and 13.6% in Hispanic/Latins (table S6). The association of these variants in Hispanics actually was much weaker compared to that in Europeans. The slight difference in R² looks like not reasonable. Could you confirm the heritability analysis for these 4 cohorts? What's the SNP h² for Hispanics and others?
- 2) The top loci from the GREY GWAS meta-analysis (Table S3) seem mainly driven by European populations. The majority of signals were flat in other populations ie. African Americans; this could be due to the small sample size. Are the beta effects concordant between these four populations?
- 3) There is no biological pathway found in this large-scale study. Previous CCT GWAS consistently pinpointed the pathway in collagen and extracellular matrix regulation (Iglesia et al 2018; Lu et al 2013). Any explanation on this?
- 4) DEPICT tissue-enrichment analysis highlighted 17 tissues. Can add some explanation/insight on this in the discussion? Does any previous evidence support it?
- 5) Line 251, it is said that "Many of the CCT-associated loci identified in this study, are also associated with glaucoma..". This is confusing as there were no pleiotropic variants found for glaucoma from the results. In the discussion, if using the previous publication to claim CCT loci associated with glaucoma, you also need to explain why the association was not observed in the current study.
- 6) What are the genomic control inflation lambdas for meta-analysis in GREY and (GREY+IGGC)? Also, what are the lambdas for 4 GWAS in GREY? This information should be given. Is there any inflation of statistics for GWAS?

Minor comments:

- 1) Line 314 "In addition to 314 those QC criteria, SNPs with genotype call rates <90% were removed, as well as SNPs with a 315 minor allele frequency <1%". As in the initial analysis, only SNVs with genotype calling rate >97% were included. Variants with call rates less than 90% have been already filtered in the first step.
- 2) For the meta-analysis of GWAS results in the GREY cohorts, is it a fixed or random effect for meta-analysis? Please specify it in the method session.
- 3) Add heterogeneity I² and p-value in table 2
- 4) Imputed variants include both SNPs and indels. In general, better call these variants as SNVs.

Reviewer #2 (Remarks to the Author):

In this multiethnic genome-wide association study (GWAS) and meta-analysis, the authors reported 74 loci from GWASs meta-analysis and 24 single nucleotide polymorphisms (SNPs) from the conditional analysis (COJO) significantly associated with central corneal thickness (CCT). Among them, 31 loci from the meta-analysis and 10 SNPs from the COJO analysis are novel. In addition, the authors calculated the SNP-based heritability of CCT and genetic explanation of CCT variability and tried to prioritize genes and SNPs for further investigation. The associations of the CCT-related SNPs with

keratoconus and primary open-angle glaucoma (POAG) were also evaluated. This study was conducted following established practices and the manuscript was well-written. It could potentially provide new insights into the genetic architecture of CCT. However, there are some important points to be clarified before being considered for publication.

Major points

1. In the methods section, it was not clear that data from which eye of the study subjects was analyzed. If both eyes were included in the analysis, the authors would need to provide more details about how the correlation between the two eyes of a subject is dealt with.
2. The authors used fixed-effect models when performing meta-analyses, ie, combining outcomes from the Genetic Epidemiology Research in Adult Health and Aging (GERA) cohort and the International Glaucoma Genetics Consortium (IGGC), and combining the results of COJO analysis from individual ethnic groups. As in fixed-effect meta-analysis, we assume there is one true effect size that underlies all the studies (in this case, study cohorts/ethnic groups) in the analysis, and that all differences in observed effects are due to sampling error. However, this assumption may not hold when combining data from populations of different ethnic backgrounds. Moreover, the authors did not report heterogeneity indexes (ie, I-square and Q statistics) in Table 2 and other meta-analytical results. Please clarify.
3. The authors reported one CCT-associated locus (rs3740685 in RAPSN) that was also associated with POAG. The two-sample Mendelian randomization (MR) analysis also showed negligible causal effects of CCT on POAG. However, some comments in the discussion were not in line with this finding. The authors emphasized significant associations of CCT-associated loci with glaucoma endophenotypes, suggesting a positive link between CCT and glaucoma.
4. It will be more informative to include an association analysis of CCT and POAG in addition to the MR analysis using the GERA cohort.

Minor points

1. Please comment on the possibility of including some potential cases with other ophthalmic conditions in the "non-cases" (GERA, N=97,375) for the keratoconus study. As also noted by the authors, UK Biobank collected self-reported glaucoma. This glaucoma population should be a heterogeneous group of glaucoma cases. Please add these points to the limitations of this study.
2. "...each individual i's CCT..."

Reviewers' comments:

Reviewer #1 (Remarks to the Author):

Choquet et al. presented data on GWAS for central corneal thickness and identified 41 novel loci. The GWAS findings of this study are significant in the field to understand CCT genetics. The data presentation is comprehensive. However, some analysis aspects need to be clarified and in general, the discussion could cover more regarding underlying biological mechanisms for CCT.

Thank you to the reviewer for the positive feedback and constructive review.

Specific comments below:

1) For GREY cohorts, top variants explain 14.2% phenotypic variation in Non-Hispanic whites, and 13.6% in Hispanic/Latinos (table S6). The association of these variants in Hispanics actually was much weaker compared to that in Europeans. The slight difference in R2 looks like not reasonable. Could you confirm the heritability analysis for these 4 cohorts? What's the SNP h2 for Hispanics and others?

While the association p-values at reported lead SNPs for Hispanic/Latinos were not as significant as those for non-Hispanic whites, the effect size estimates were quite similar, resulting similar estimates of phenotypic variation explained. To fully address this concern, we compared the betas for the 98 top CCT-associated SNPs (74 from the combined meta-analysis + 24 from the COJO analysis) between GERA non-Hispanic whites and GERA Hispanic/Latinos using a Z-test. After Bonferroni correction (0.05/98 SNPs tested), we found that the betas were not significantly different between GERA non-Hispanic whites and GERA Hispanic/Latinos. We have added a Supplementary Table (Supp. Table 6) to present these comparison results.

Further, in addition to present the array-heritability estimate of CCT in GERA non-Hispanic whites, we now assessed the array-heritability estimate of CCT in the other three GERA ethnic groups (Hispanic/Latinos, East Asians, and African American) using GCTA tool, and completed the Supplementary Table 7, as follows:

Ethnic group	Array h^2 (SE)	GCTA-GREML power (%)	Variance explained Adjusted R^2 (%)
Non-Hispanic whites	0.425 (0.033)	100%	14.2
Hispanic/Latinos	0.113 (0.401)	46%	13.6
East Asians	0.113 (0.395)	54%	8.9
African Americans	0.000001 (0.416)	17.7%	10.1

However, the array-heritability estimate of CCT in GERA Hispanic/Latinos, East Asians and African Americans should be considered cautiously as these samples are limited in size and, consequently, we do not have sufficient statistical power for accurate estimates. According to the GCTA software guidelines: "It is not recommended to run a GCTA-GREML analysis in a small sample. When the sample size is small, the sampling variance (standard error squared) of the estimate is large (see GCTA-GREML power calculator), so the estimate of SNP-heritability (h^2 -SNP) will fluctuate a lot and could even hit the boundary (0 or 1). Therefore, when the sample size is small, it is not surprising to observe an estimate of SNP-heritability being 0 or 1 (with a large standard error)."

We estimated that in GERA non-Hispanic whites, we have 100% power of detecting $h^2 > 0$ for the given the user-specified type I error rate and the SNP-heritability assumed in the population (~43%). In contrast, we have only 54% power in GERA East Asians, 46% power in GERA Hispanic/Latinos, and 17.7% power in GERA African American. For this reason, in our main manuscript, we reported only the array-heritability estimate for GERA non-Hispanic whites $h^2 = 0.425$ (0.033)).

2) The top loci from the GREY GWAS meta-analysis (Table S3) seem mainly driven by European populations. The majority of signals were flat in other populations ie. African Americans; this could be due to the small sample size. Are the beta effects concordant between these four populations?

We have now added a column in Supplementary Table 3 entitled “Directions of effect (EUR/EAS/AFR/LAT)” to indicate the direction of effect for the 28 CCT-associated SNPs (identified in the GERA meta-analysis) across the 4 GERA ethnic groups: GERA non-Hispanic whites, East Asians, African American, Hispanic/Latinos. Most SNPs (78.6%) showed consistent direction of effects across all ethnic groups.

3) There is no biological pathway found in this large-scale study. Previous CCT GWAS consistently pinpointed the pathway in collagen and extracellular matrix regulation (Iglesias et al 2018; Lu et al 2013). Any explanation on this?

In our current study we presented biological pathway annotations and prioritization results from the DEPICT integrative tool. In the Iglesias’ study, authors presented biological pathway annotations and prioritization results based on the VEGAS2 software. This may explain the differences in biological pathway detection between the 2 studies, although our DEPICT gene-set enrichment analysis also detected nominal evidence for previously reported gene-sets, including those involved in the regulation of extracellular matrix organization, and collagen formation.

For consistency and comparison between the Iglesias’ study and our current study, we now conducted a VEGAS2-pathway analysis on the combined (GERA + IGGC) meta-analysis results and reported the findings in a Supplementary Table (Supplementary Table 13). Using a 10 kb window in the VEGAS2 computation, we found that 25 pathways/gene-sets were significantly enriched after correcting for multiple testing ($P < 5.14 \times 10^{-6}$), compared to 23 pathways/gene-sets identified in the Iglesias’ study. Similarly, most of these pathways/gene-sets contribute to the function of the extracellular matrix and collagen. In addition, we identified gene-sets related to head/face morphogenesis and development.

4) DEPICT tissue-enrichment analysis highlighted 17 tissues. Can add some explanation/insight on this in the discussion? Does any previous evidence support it?

As suggested by the reviewer, we have now added some explanation/insight on the DEPICT tissue-enrichment analysis results in the Discussion, as follows:

“Similarly, our study revealed potential biological pathways and relevant tissues involved in CCT variation that are pertained to the function of fibroblast and collagen-rich tissues, as well as the regulation of epithelial to mesenchymal transition, and the organization of the extracellular matrix, consistent with previous works. Functional follow-up experiments in cell lines or animal models may confirm the involvement of these genes and biological pathways in CCT variation and reveal the underlying mechanisms of CCT-related vision disorders.”

5) Line 251, it is said that “Many of the CCT-associated loci identified in this study, are also associated with glaucoma..”. This is confusing as there were no pleiotropic variants found for glaucoma from the results. In the discussion, if using the previous publication to claim CCT loci associated with glaucoma, you also need to explain why the association was not observed in the current study.

We have now rephrased this sentence in the Discussion as below: “Many of the CCT-associated loci identified in this study, are also associated with ~~glaucoma endophenotypes and~~ other eye conditions”.

“Other eye conditions” include: myopia, age-related macular degeneration, eye color, corneal/refractive astigmatism, strabismus, macular thickness, primary congenital glaucoma, and exfoliation syndrome.

6) What are the genomic control inflation lambdas for meta-analysis in GRAE and (GRE+IGGC)? Also, what are the lambdas for 4 GWAS in GRE? This information should be given. Is there any inflation of statistics for GWAS?

We now provide QQ-plots and report the corresponding genomic control inflation lambdas for the 4 GERA ethnic groups GWAS of CCT (Supplementary Figures 1-4) and meta-analysis GWAS across these 4 GERA ethnic groups (Supplementary Figure 5) in addition to Supplementary Figure 6 (QQ plot and genomic inflation factors (λ) observed for the combined (GERA+IGGC) multiethnic meta-analysis of CCT). The genomic control inflation lambdas for each analysis are reported in the following table, and are in line with expectations for a study of this size and our previous experience:

GERA Ethnic group	Lambda
Non-Hispanic white	1.059

Hispanic/Latino	0.987
East Asian	0.987
African American	0.994
GERA meta-analysis	1.059

Minor comments:

1) Line 314 “In addition to those QC criteria, SNPs with genotype call rates <90% were removed, as well as SNPs with a minor allele frequency <1%”. As in the initial analysis, only SNVs with genotype calling rate >97% were included. Variants with call rates less than 90% have been already filtered in the first step.

We thank the reviewer for drawing our attention to this sentence related to the description of initial genotype QC. We have now rephrased it in the Methods section as follows:

“Genotype QC (quality control) procedures were performed on an array-wise basis⁶⁰, as follows: SNPs with initial genotyping call rate $\geq 97\%$, allele frequency difference (≤ 0.15) between males and females for autosomal markers, and genotype concordance rate (> 0.75) across duplicate samples were included. About 94% of samples and more than 98% of genetic markers assayed passed QC procedures. In addition to those QC criteria, SNPs with a minor allele frequency $< 1\%$ were removed.”

2) For the meta-analysis of GWAS results in the GREY cohorts, is it a fixed or random effect for meta-analysis? Please specify it in the method session.

The meta-analysis results reported in Table 2 are fixed-effects. We have now specified in the Methods section as follows: “**Meta-analysis.** To combine the study results of Iglesias et al. with our GERA meta-analysis, we conducted a fixed-effect meta-analysis. Heterogeneity index, I^2 (0–100%) and P -value for Cochran’s Q statistic among studies were assessed.”

3) Add heterogeneity I^2 and p -value in table 2

We have now added a column reporting Heterogeneity index, I^2 (0–100%) and P -value for Cochran’s Q statistic among studies in Table 2.

4) Imputed variants include both SNPs and indels. In general, better call these variants as SNVs.

We have now replaced “SNPs” by “SNVs” in some places in the manuscript (e.g. at the end of the Introduction section, in Table 2, etc.).

Reviewer #2 (Remarks to the Author):

In this multiethnic genome-wide association study (GWAS) and meta-analysis, the authors reported 74 loci from GWASs meta-analysis and 24 single nucleotide polymorphisms (SNPs) from the conditional analysis (COJO) significantly associated with central corneal thickness (CCT). Among them, 31 loci from the meta-analysis and 10 SNPs from the COJO analysis are novel. In addition, the authors calculated the SNP-based heritability of CCT and genetic explanation of CCT variability and tried to prioritize genes and SNPs for further investigation. The associations of the CCT-related SNPs with keratoconus and primary open-angle glaucoma (POAG) were also evaluated. This study was conducted following established practices and the manuscript was well-written. It could potentially provide new insights into the genetic architecture of CCT. However, there are some important points to be clarified before being considered for publication.

We thank the reviewer for the helpful comments.

Major points

1. In the methods section, it was not clear that data from which eye of the study subjects was analyzed. If both eyes were included in the analysis, the authors would need to provide more details about how the correlation between the two eyes of a subject is dealt with.

We have now clarified the fact that CCT measurements from both eyes were used in the current study and provided more details in the Methods section, as follows:

“For the current study, 18,129 GERA participants from four ethnic groups who had at least one recorded CCT measurement on both eyes during the same visit between June 2014 and January 2018 were included. (...) **CCT measurement.** CCT was measured in GERA using the DGH-550 or DGH-55 ultrasonic (contact) pachymeter (DGH Technology Inc.; Exton, PA), or a non-contact optical biometer (Lenstar LS900, Haag-Streit, Köniz, Switzerland), and recorded for both eyes in the electronic health records. Patients with single eye measurements were removed. (...) the mean standardized CCT of both eyes and standard deviation (sd) were assessed for each patient. Outliers (N=9) defined by large left-right differences (i.e., beyond 4 sd of the overall standardized distribution of left-right differences) were removed.”

We have also added a Supplementary Figure (Supp. Figure 14) showing the distribution of CCT measures in both the left and the right eye of each individual for all the measurements used in the current study.

2. The authors used fixed-effect models when performing meta-analyses, ie, combining outcomes from the Genetic Epidemiology Research in Adult Health and Aging (GERA) cohort and the International Glaucoma Genetics Consortium (IGGC), and combining the results of COJO analysis from individual ethnic groups. As in fixed-effect meta-analysis, we assume there is one true effect size that underlies all the studies (in this case, study cohorts/ethnic groups) in the analysis, and that all differences in observed effects are due to sampling error. However, this assumption may not hold when combining data from populations of different ethnic backgrounds. Moreover, the authors did not report heterogeneity indexes (ie, I-square and Q statistics) in Table 2 and other meta-analytical results. Please clarify.

As also suggested by reviewer 1, we have now added a column reporting the Heterogeneity index, I^2 (0–100%) and P -value for the Cochrane’s Q statistic among studies in Table 2 and Supplementary Table 4.

3. The authors reported one CCT-associated locus (rs3740685 in RAPSN) that was also associated with POAG. The twosample Mendelian randomization (MR) analysis also showed negligible causal effects of CCT on POAG. However, some comments in the discussion were not in line with this finding. The authors emphasized significant associations of CCT associated loci with glaucoma endophenotypes, suggesting a positive link between CCT and glaucoma.

As also highlighted by reviewer 1, we have now clarified the text in the Discussion, as follows: “Many of the CCT-associated loci identified in this study, are also associated with ~~glaucoma endophenotypes~~ and other eye conditions”.

‘Other eye conditions’ include: myopia, age-related macular degeneration, eye color, corneal/refractive astigmatism, strabismus, macular thickness, primary congenital glaucoma, and exfoliation syndrome.

4. It will be more informative to include an association analysis of CCT and POAG in addition to the MR analysis using the GERA cohort.

We have now added a paragraph showing the association between CCT and POAG in GERA in the Results section, before the MR analysis, as follows:

“In GERA, POAG status was significantly associated with lower CCT after adjusting for age, sex, ethnic group, and CCT measurement type (beta=-12.57 and $P=2.33 \times 10^{-66}$) (Supplementary Table 16). This significant association between POAG and lower CCT was true for all the GERA ethnic groups.”

Minor points

1. Please comment on the possibility of including some potential cases with other ophthalmic conditions in the "noncases" (GERA, N=97,375) for the keratoconus study. As also noted by the authors, UK Biobank collected self-reported glaucoma. This glaucoma population should be a heterogeneous group of glaucoma cases. Please add these points to the limitations of this study.

As suggested by the Reviewer, we have now emphasized these points as potential limitations of the study in the Discussion as follows : “We recognize several potential limitations of our study. First, the ‘non-cases’ of the keratoconus GERA study may include some potential cases with other ophthalmic conditions, which may result in underestimates of the effects of individual CCT-associated SNPs if those conditions are also associated with the risk of keratoconus. Second,

glaucoma diagnoses in UKB were based on self-reported data, and the subtypes of glaucoma were unspecified, which may result in underestimates of the effects of individual SNPs due to phenotype misclassification. However, our glaucoma results were consistent across GERA and UKB.”

2. "...each individual i's CCT..."

In the Methods section, under “Genetic ancestry analysis”, we have made some edits in the text to be consistent with the radial kernel density equation, as follows:

“Genetic ancestry analysis. A full description of the ancestry analyses in GERA is provided in Banda et al.¹⁵. The CCT distribution by the ancestry PCs for each GERA ethnic groups is illustrated on **Figure 1**. To create these plots, a smoothed distribution of each individual *i*'s CCT phenotype was created using a radial kernel density estimate weighted on the distance to each other *j*th individual, as follows:

$$\sum_j \phi(\{d(i, j) / \max_{i', j'} [d(i', j')] \times k\})$$

REVIEWERS' COMMENTS:

Reviewer #1 (Remarks to the Author):

The comments have been addressed in the revision

Reviewer #2 (Remarks to the Author):

The authors have addressed my comments. Thanks for the revision.